Development and validation of a mitotic catastrophe-related genes prognostic model for breast cancer

Wang Shuai
Zi Haoyi
Li Mengxuan
Kong Jing
Fan Cong
Bai Yujie
Sun Jianing
Wang Ting ting_w100@126.com
The First Affiliated Hospital of Air Force Medical University, Department of Thyroid, Breast and Vascular Surgery , Xi’an , Shaanxi , China
Zhao Min
Electronic publication date: 2024 Sep 20
Publication date: 2024
Volume: 12
Electronic Location ID: e18075
Received 2024 May 16; Accepted 2024 Aug 19
Copyright: ©2024 Wang et al.
Copyright year: 2024
Copyright holder: Wang et al.
License: This is an open access article distributed under the terms of the Creative Commons Attribution License, which permits unrestricted use, distribution, reproduction and adaptation in any medium and for any purpose provided that it is properly attributed. For attribution, the original author(s), title, publication source (PeerJ) and either DOI or URL of the article must be cited.
License URL: https://creativecommons.org/licenses/by/4.0/

Keywords: Breast cancer, Mitotic catastrophe, Immune microenvironment, Bioinformatics

Funding: Shaanxi Province Natural Science Basic Research 2021JZ-29 2023-JC-QN-0965 Cultivation Boost Project of Xijing Hospital XJZT24LY09 This work was supported by the Shaanxi Province Natural Science Basic Research (No. 2021JZ-29), and the Shaanxi Province Natural Science Basic Research (No. 2023-JC-QN-0965) and the Cultivation Boost Project of Xijing Hospital (No. XJZT24LY09). The funders had no role in study design, data collection and analysis, decision to publish, or preparation of the manuscript.

==============================
Background

Breast cancer has become the most common malignant tumor in women worldwide. Mitotic catastrophe (MC) is a way of cell death that plays an important role in the development of tumors. However, the exact relationship between MC-related genes (MCRGs) and the development of breast cancer is still unclear, and further research is needed to elucidate this complexity.

Methods

Transcriptome data and clinical data of breast cancer were downloaded from the Cancer Genome Atlas (TCGA) database and the Gene Expression Omnibus (GEO) database. We identified differential expression of MCRGs by comparing tumor tissue with normal tissue. Subsequently, we used COX regression analysis and LASSO regression analysis to construct the prognosis risk model of MCRGs. Kaplan–Meier survival curve and receiver operating characteristic (ROC) curve were used to evaluate the predictive ability of prognostic model. Moreover, the clinical relevance, gene set enrichment analysis (GSEA), immune landscape, tumor mutation burden (TMB), and immunotherapy and drug sensitivity analysis between high-risk and low-risk groups were systematically investigated. Finally, we validated the expression levels of genes involved in constructing the prognostic model through real-time quantitative polymerase chain reaction (RT-qPCR) at the cellular and tissue levels.

Results

We identified 12 prognostic associated MCRGs, four of which were selected to construct prognostic model. The Kaplan-Meier analysis suggested that patients in the high-risk group had a shorter overall survival (OS). The Cox regression analysis and ROC analysis indicated that risk model had independent and excellent ability in predicting prognosis of breast cancer patients. Mechanistically, a remarkable difference was observed in clinical relevance, GSEA, immune landscape, TMB, immunotherapy response, and drug sensitivity analysis. RT-qPCR results showed that genes involved in constructing the prognostic model showed significant abnormal expressions and the expression change trends were consistent with the bioinformatics results.

Conclusions

We established a prognosis risk model based on four MCRGs that had the ability to predict clinical prognosis and immune landscape, proposing potential therapeutic targets for breast cancer.

Introduction

Breast cancer is the most common malignant tumor in women. The GLOBOCAN2022 report (Bray et al., 2024) shows that there are 2.32 million new cases of breast cancer globally, with over 660,000 deaths. With the development of early diagnosis and treatment strategies, the prognosis of breast cancer patients has significantly improved, with a 5-year survival rate of up to 90% (Yersal & Barutca, 2014). However, breast cancer is a highly heterogeneous tumor, and breast cancer with the same clinical stage and pathological type may have completely different prognoses (Takada & Toi, 2020). In addition, due to some patients developing resistance to chemotherapy, endocrine therapy, or targeted therapy (Hanker, Sudhan & Arteaga, 2020; Mehraj et al., 2021), the overall survival (OS) of breast cancer patients is still not ideal. Therefore, it is very important to find reliable prognostic indicators and effective treatment targets.

Mitosis is the process by which eukaryotic cells produce somatic cells, and it occurs periodically. The mitotic cycle is divided into four stages: G1, S, G2, and M. When there are abnormalities in the cell cycle checkpoints or damage to the spindle apparatus, it can lead to cell death, a phenomenon known as mitotic catastrophe (MC) (Weaver & Cleveland, 2005). MC is a form of cell death that plays an important role in the development of tumors (Bai et al., 2023). MC can inhibit cell proliferation and induce cell death, including apoptosis, autophagy, and necrosis (Sazonova et al., 2021). MC affects immune cells in the tumor microenvironment to regulate immune responses, and is associated with tumor immune escape and the efficacy of immunotherapy (Mao et al., 2024). In addition, MC is closely related to the resistance of tumor cells to drugs and radiotherapy. Studies have shown that inducing MC in glioblastoma and lung cancer cells can enhance the sensitivity of tumor cells to radiation (Oike et al., 2014; Tandle et al., 2013), providing a new approach for clinical radiotherapy of tumors. Additionally, the occurrence of MC in multiple myeloma cells can overcome resistance to L-phenylalanine nitrogen mustard (Hu et al., 2022), suggesting that MC has potential value in overcoming drug resistance in tumor cells. With further exploration of the phenomenon and mechanisms of MC, it is hoped that MC could become a new approach for cancer treatment. MC-related genes (MCRGs) affect the prognosis of tumor patients. In prostate cancer, the expression level of Cyclin K is associated with the recurrence-free survival of patients, and it can serve as a biomarker for patient prognosis (Schecher et al., 2017). In colorectal adenocarcinoma, researchers have constructed a prognostic model based on 5 MCRGs, which can serve as a reliable prognostic biomarker for patients (Liu et al., 2024). However, to date, there has been no research on constructing a prognostic model based on MCRGs in breast cancer.

This study utilized public databases for bioinformatics analysis, screened MCRGs affecting the prognosis of breast cancer patients, and constructed a prognostic risk score model. At the same time, we preliminarily explored the relationship between risk score model and immune infiltration, providing a new direction for future basic research. We have drawn a flowchart to systematically describe our research (Fig. 1).

Figure 1 The flow diagram of this study.

Materials & Methods

Data acquisition

The gene expression data, gene mutation information, and clinical data of breast cancer patients were retrieved from The Cancer Genome Atlas (TCGA) (https://portal.gdc.cancer.gov/) database (TCGA-BRCA) (Love, Huber & Anders, 2014) and the Gene Expression Omnibus (GEO) (https://www.ncbi.nlm.nih.gov/geo/) database (GSE20685) (Kao et al., 2011). TCGA-BRCA served as the training set, while GSE20685 served as the validation set. The “sva” package (Leek et al., 2012) in R software was used to correct batch effects between different datasets. Immunohistochemical data were downloaded from the Human Protein Atlas (HPA) (https://www.proteinatlas.org/) (Pontén, Jirström & Uhlen, 2008) database. The MC gene set, including 900 MCRGs, was downloaded from the GENECARDS (https://www.genecards.org/) (Rebhan et al., 1997) database (Table S1).

Differential expression analysis

We used the “limma” package in R software and Wilcoxon test to perform differential analysis on normal samples and tumor samples in the training set, and obtained MCRGs differentially expressed in breast cancer. The filtering criteria were —logFC—>1 and FDR < 0.05.

GO and KEGG enrichment analysis

The “clusterprofiler” package (Yu et al., 2012) in R software was used to perform Gene ontology (GO) and Kyoto Encyclopedia of Genes and Genomes (KEGG) enrichment analysis on MCRGs in differential expression, evaluating the pathways and functions involved in differential genes. GO analysis had three types, which were molecular function (MF), biological process (BP) and cellular component (CC).

Construction and validation of the prognosis risk model

First, the R software “survival” package was used to perform univariate Cox regression analysis in the training set, screening out genes that may be related to patient prognosis. Then, we used the R software “glmnet” package to conduct LASSO regression analysis to eliminate redundant genes and perform multivariate Cox regression analysis, ultimately selecting prognosis-related genes. Finally, we built a prognostic risk model based on the regression coefficients and expression levels of each gene. The risk score = β1 ×Gene1EXP + β2 ×Gene2EXP + ...... + βn ×GenenEXP, where β is the regression coefficient of the corresponding gene, and GeneEXP is the expression level of the corresponding gene. We calculated the risk score for each breast cancer patient, and divided patients into high-risk and low-risk groups based on the median risk score. The R software “survival” package was used to plot Kaplan–Meier survival curves, comparing the OS of high-risk and low-risk groups; the “survminer” package was used to plot receiver operating characteristic (ROC) curves, evaluating the predictive performance of the model based on the area under the curve (AUC); the “pheatmap” package was used to plot heatmaps of gene expression and survival status in the prognostic model. The validation set was used to evaluate the effectiveness of the model using the same coefficients and median values as the training set.

Construction and evaluation of a predictive nomogram

We used risk score as a variable, combined with other clinical parameters (age, T stage, N stage, M stage) for univariate and multivariate Cox regression analysis to determine independent prognostic factors for breast cancer patients. The R software packages “rms” and “regplot” were used to construct a predictive nomogram for predicting 1, 3, and 5-year OS of breast cancer patients. Calibration plots and ROC curves were used to demonstrate the predictive ability of the predictive nomogram.

Risk score and clinicopathological characteristics

We divided breast cancer patients in training set into subgroups based on clinicopathological characteristics. Then, the associations between risk score and clinicopathological characteristics were identified, and the results were displayed in a box plot.

GSEA

Gene set enrichment analysis (GSEA) is a computational method that determines if a set of genes, defined a priori based on known biological pathways or functional annotations, is significantly enriched in a list of differentially expressed genes. In this study, GSEA of the high-risk and low-risk groups was created by the desktop application of GSEA 4.3.2. The gene sets from the “c2.cp.kegg.v7.2.symbols.gmt” collection in the Molecular Signatures Database (https://www.gseamsigdb.org/) (Liberzon et al., 2015) were used for GSEA (Subramanian et al., 2005). To assess the significance of the enrichment scores obtained from GSEA, 1,000 gene set permutations were conducted to obtain a normalized enrichment score for each analysis. Pathways were considered statistically enriched at the cut-off point of P < 0.05 and FDR < 0.25.

Tumor immune microenvironment

The ESTIMATE algorithm (Yoshihara et al., 2013) was used to calculate the immune infiltration score of breast cancer samples; the CIBERSORT algorithm (Chen et al., 2018) was used to assess the correlation between immune cells and risk scores; the ssGSEA algorithm and the R software “GSVA” package were used to evaluate the differences in immune function between high-risk and low-risk groups. We obtained 47 immune checkpoint genes from the previous study (Danilova et al., 2019), and used the R software “ggpubr” package to compare the activation status of immune checkpoints between high-risk and low-risk groups.

Mutational landscape

Tumor mutation burden (TMB) refers to the number of somatic mutations in a tumor genome after excluding germline mutations, defined as the total number of somatic gene coding errors, base substitutions, gene insertions, or deletions detected per million bases. TMB is considered a biological marker for measuring the level of tumor mutations (Chan et al., 2019). The R software package “maftools” (Mayakonda et al., 2018) was used to calculate the TMB for each sample, and waterfall plots were generated to display the mutation landscape of high-risk and low-risk groups of breast cancer patients.

Immunotherapy response analysis and drug sensitivity analysis

The Tumor Immune Dysfunction and Exclusion (TIDE) (http://tide.dfci.harvard.edu/) (Jiang et al., 2018) database was employed to assess the response to immune therapy. The R software package “oncoPredict” (Maeser, Gruener & Huang, 2021) was used to calculate the half inhibitory concentration (IC50) of commonly used chemotherapy drugs to evaluate the predictive ability of MCRGs on drug treatment response. Then, we used the Wilcoxon test to compare the differences in IC50 between high-risk and low-risk groups, and displayed the results in a box plot.

Cell culture and tissue collection

Human normal mammary epithelial cells MCF-10A, human breast cancer cells MDA-MB-231 and MCF-7 were purchased from American Type Culture Collection. These cells were cultured in DMEM or RPMI-1640 (Gibco, Waltham, MA, USA). All cells were grown at 37 °C with 5% CO2 in 10% fetal bovine serum (FBS) from Gibco BRL in the United States. We obtained 10 pairs of breast cancer tissues and the paired normal adjacent tissues from patients without preoperative chemotherapy, endocrine therapy, or radiotherapy who had undergone tumor resection at The First Affiliated Hospital of Air Force Medical University. Our hospital’s Institutional Ethical Board gave the study its approval (KY20232266-C-1). We obtained written consent from the patients. All methods were performed in accordance with the relevant guidelines and regulations.

Real-time quantitative polymerase chain reaction

Total RNA was extracted using SPARKeasy Improved Tissue/Cell RNA Kit (Sparkjade Biotech Co., Ltd., Shandong, China) following the manufacturer’s instructions. cDNA was synthesized using SPARKscript II All-in-one RT SuperMix for qPCR (Sparkjade). Real-time quantitative polymerase chain reaction (RT-qPCR) was performed using 2 ×SYBR Green qPCR Mix (Sparkjade). The internal controls were β-actin. Tsingke Biotech (Xi’an, China) designed all primers, and detailed primer sequences were presented in Table S2. Gene expression levels were quantitatively calculated by the 2−ΔΔCt method. All samples were tested in triplicate.

Statistical analysis

Data analysis and graph plotting were conducted using R software version 4.2.2. Student’s t-test or Wilcoxon test was used for analyzing continuous variable data, while Chi-square test was used for analyzing categorical data. Kaplan–Meier analysis and log-rank test were used to compare survival differences. A P-value <0.05 was considered statistically significant. ∗P < 0.05, ∗∗P < 0.01, ∗∗∗P < 0.001.

Results

Differential expression analysis and enrichment analysis

We conducted differential analysis of MCRGs in normal and tumor samples in TCGA-BRCA, and identified 194 genes that were differentially expressed in breast cancer, including 127 upregulated genes and 67 downregulated genes (Fig. 2A). GO analysis was performed on differentially expressed genes, and the results showed that the significantly enriched MF of these differentially expressed genes mainly included tubulin binding, protein serine/threonine kinase activity, and protein serine kinase activity; BP mainly included mitotic cell cycle phase transition, regulation of cell cycle phase transition, and organelle fission; CC mainly included condensed chromosome, chromosomal region, and spindle (Fig. 2B). Further KEGG analysis revealed that the differentially expressed genes were mainly involved in nuclear division, regulation of mitotic cell cycle phase transition, chromosome segregation, negative regulation of cell cycle, and cell cycle checkpoint signaling (Fig. 2C).

Figure 2 Characterization of differentially expressed MCRGs in breast cancer.

(A) Volcano plot of differentially expressed MCRGs. (B) GO analyses of the 194 differentially expressed MCRGs. (C) KEGG analyses of the 194 differentially expressed MCRGs.

Construction and validation of MCRGs-related prognostic model

In the training set, we first screened out 12 genes that may be related to patient prognosis through univariate Cox regression analysis (Fig. 3A). Then we conducted LASSO regression analysis to eliminate redundant genes and performed multivariate Cox regression analysis, screening out four prognosis-related genes (PLK1, S100B, IRS2 and IFNG) (Figs. 3B–3D). The expression levels of PLK1 and IFNG are upregulated, while the expression levels of S100B and IRS2 are downregulated. Finally, we constructed a prognostic risk model based on the regression coefficients and expression levels of each gene. We calculated the risk score for each breast cancer patient in the training set and divided the patients into high-risk and low-risk groups based on the median risk score. We plotted gene expression heatmaps and survival status plots. The survival status analysis results showed that as the risk score increased, the proportion of deceased patients significantly increased (Figs. 4A and 4B). Kaplan–Meier survival analysis results showed that compared to the high-risk group, patients in the low-risk group had a longer OS (Fig. 4C). The ROC curve results showed that the model predicted the AUC of 1, 3, and 5-year OS for breast cancer patients to be 0.711, 0.660, and 0.632 respectively, indicating that the model had predictive ability (Fig. 4D). We used the same coefficients as the training set to calculate the risk score of each breast cancer patient in the validation set, and divided the patients into high-risk and low-risk groups based on the same median values as the training set. Survival status analysis, survival analysis, and ROC curve analysis in the validation set all indicated that the model had high robustness (Figs. 4E–4H).

Figure 3 Identification of candidate MCRGs for generating risk model of breast cancer.

(A) Forest plot of univariate Cox analysis showing the 12 MCRGs significantly associated with OS in breast cancer patients. (B) The LASSO coefficient profile of 12 differentially expressed MCRGs. (C) The tenfold cross-validation for variable selection in the LASSO model. (D) The role of 4 model genes.

Figure 4 Construction and validation of prognostic risk profiles in breast cancer patients with four model genes.

(A and E) Heat map including four model genes in training set and validation set. (B and F) Risk plot distribution and survival status of patients in training set and validation set. (C and G) Kaplan–Meier survival curves of OS for patients between low-risk and high-risk groups in training set and validation set. (D and H) ROC curves for predictive performance of the risk model in training set and validation set.

Development and evaluation of MCRGs-correlated clinicopathologic nomogram

The results of univariate and multivariate Cox regression analysis showed that the risk score was an independent prognostic factor for breast cancer patients in both the training set and validation set (Figs. 5A and 5B). A nomogram was constructed combining clinical parameters and risk score (Fig. 5C). The calibration curve showed a high consistency between the prognostic model and actual observations, indicating good predictive performance of the nomogram (Fig. 5D). The ROC curve showed that the nomogram model combining clinical parameters and risk score predicted the AUC of 1, 3, and 5-year OS for breast cancer patients to be 0.889, 0.739, and 0.734 respectively, indicating that the nomogram model had a high predictive ability for the prognosis of breast cancer patients (Fig. 5E).

Figure 5 Development and evaluation of predictive nomogram.

(A and B) Univariate and multivariate Cox analyses of clinical factors and risk score with OS in training set and validation set. (C) Nomogram predicting 1, 3 and 5-year survival rate of breast cancer patients. (D) The calibration curves for 1, 3 and 5-year OS in training set. (E) ROC curves for predictive performance of the nomogram model in training set.

Clinical relevance of risk model and GSEA

Clinical relevance analysis found significant differences in risk scores among different age groups, T stages, and N stages (Fig. 6A). GSEA aims to elucidate the potential regulatory mechanisms underlying the differences between high-risk and low-risk groups. The results showed that the high-risk group was significantly associated with base excision repair, cell cycle, DNA replication, and oxidative phosphorylation (Fig. 6B); while the low-risk group was significantly associated with chemokine signaling pathway, JAK-STAT signaling pathway, natural killer cell mediated cytotoxicity, and T cell receptor signaling pathway (Fig. 6C).

Figure 6 Clinical correlation analysis and GSEA.

(A) Correlation between the risk score and clinical characteristics (age; T stage; N stage; M stage). (B and C) Enrichment plots from GSEA analysis in the high-risk and low-risk groups.

Immune landscape

In order to further explore the relationship between risk score and tumor immune microenvironment, we compared the immune infiltration scores between high-risk and low-risk groups, and found that the Stromal score, Immune score, and ESTIMATE score were all different (Fig. 7A). The ssGSEA algorithm was used to calculate the enrichment scores of various immune cell subtypes, related functions, or pathways. The results showed that the infiltration of CD8+ T cells, M0 macrophages, and M2 macrophages between the high-risk and low-risk groups had significant differences (Fig. 7B). In terms of immune function, helper T cells, tumor-infiltrating lymphocytes, and type II interferon response were more active in the low-risk group (Fig. 7C). Immune checkpoint analysis showed that the majority of immune checkpoints were more activated in the low-risk group (Fig. 7D).

Figure 7 Risk score and tumor immune microenvironment.

(A) Comparison of immune infiltration scores (including Stromal score, Immune score, and ESTIMATE score) between low-risk and high-risk groups. (B) Comparison of tumor-infiltrating immune cells between low-risk and high-risk groups. (C) Comparison of immune-function score between low-risk and high-risk groups. (D) Comparison of immune checkpoint genes between low-risk and high-risk groups.

TMB analysis

We calculated the TMB of breast cancer patients and displayed the top 20 genes based on mutation frequency in a waterfall plot (Figs. 8A and 8B). We found that three out of the four MCRGs involved in building the prognostic model had mutated, which were PLK1, S100B, and IFNG. In addition, there was a significant difference in TMB levels between the high-risk and low-risk groups (Fig. 8C). Survival analysis showed that TMB affected patient prognosis, with higher levels of TMB being associated with poorer OS (Figs. 8D and 8E).

Figure 8 Mutation status.

(A and B) The top 20 genes according to mutation frequency in low-risk and high-risk groups, respectively. (C) Comparison of TMB levels between low-risk and high-risk groups. (D and E) Kaplan–Meier curves of OS for patients in the high-TMB and low-TMB groups.

Immunotherapy and drug sensitivity analysis

In order to determine if patients with different risk patterns have different responses to immunotherapy, we conducted TIDE analysis. According to the research findings, the high-risk group had lower TIDE scores, suggesting that they may have a better response to immunotherapy (Fig. 9A). In addition, we compared the sensitivity of the high-risk and low-risk groups to common chemotherapy drugs to determine potential treatment options. The results showed that patients in the low-risk group were sensitive to chemotherapy drugs such as 5-Fluorouracil, Palbociclib, and Fludarabine (Figs. 9B–9D). While patients in the high-risk group were sensitive to chemotherapy drugs such as Sapitinib, Acetalax, Dihydrorotenone, and OSI-027 (Figs. 9E–9H). This will provide guidance for clinical selection of the most suitable drugs.

Figure 9 Immunotherapy and drug sensitivity analysis.

(A) Comparison of responses to immunotherapy between low-risk and high-risk groups. (B–H) Comparison of the 5-Fluorouracil, Palbociclib, Fludarabine, Sapitinib, Acetalax, Dihydrorotenone, and OSI-027 between low-risk and high-risk groups.

Validation of MCRGs expression in risk model

In order to verify the expression of the 4 genes involved in building the prognosis model, we downloaded immunohistochemical staining images from the HPA database. PLK1, IFNG, S100B, and IRS2 were expressed at significantly different levels between breast cancer and normal breast tissues (Figs. 10A–10D). Then, we conducted RT-qPCR assay on 10 pairs of breast cancer tissues and the paired adjacent normal tissues. The expression levels of PLK1 and IFNG in breast cancer tissues were significantly higher than those in the paired adjacent normal tissues. The expression levels of S100B and IRS2 in breast cancer tissues were significantly lower than those in the paired adjacent normal tissues (Figs. 11A–11D). We also validated the expression of these 4 genes in human normal mammary epithelial cells MCF-10A and human breast cancer cells MCF-7 and MDA-MB-231 through RT-qPCR. The results showed that the expression of PLK1 and IFNG in human breast cancer cells were significantly higher than in human normal mammary epithelial cells; while the expression of S100B and IRS2 were significantly lower than in human normal mammary epithelial cells (Figs. 11E–11H). These results supported our hypothesis and provided solid evidence for the rationality of selecting these four genes to build the prognosis model.

Figure 10 The protein expression of four model genes in breast cancer and normal tissues in the HPA database.

Figure 11 Experimental verification of four model genes expression in breast cancer.

(A–D) Expression of four model genes in 10 paired breast cancer tissues and normal tissues was evaluated by RT-qPCR. (E–H) Expression of four model genes in a human normal mammary epithelial cell line MCF-10A and human breast cancer cell lines through RT-qPCR. *P < 0.05, **P < 0.01, ***P < 0.001, ****P < 0.0001.

Discussion

Breast cancer is the most common malignant tumor in women worldwide and the incidence is increasing. Treatment options for breast cancer include surgery, radiation therapy, and drug therapy. Early invasive breast cancer and ductal carcinoma in situ are usually treated primarily with surgery (Hassett et al., 2020), with radiation therapy as an adjuvant therapy to reduce the risk of recurrence after surgery (Shah et al., 2020). Drug therapy is used for systemic treatment of breast cancer (Korde et al., 2021), and the treatment plan depends on the molecular subtype of breast cancer. Approximately 70% of breast cancer patients are estrogen receptor-positive or progesterone receptor-positive, commonly treated with endocrine therapy (Wolff et al., 2007); patients with overexpression of human epidermal growth factor receptor 2 (HER2) account for 15% to 20%, commonly treated with targeted therapy (Romond et al., 2005; Slamon et al., 2001); triple-negative breast cancer still lacks targeted therapy methods, and traditional chemotherapy remains the main treatment option (Duffy, McGowan & Crown, 2012). Recurrence and metastasis are the biggest challenges in breast cancer treatment currently, so finding new diagnostic and treatment targets, developing new treatment methods for breast cancer, is an urgent issue that needs to be addressed in breast cancer treatment.

MC is a strategy to eliminate non-functional cells undergoing mitosis in higher eukaryotes, driven by complex and poorly understood signal cascades. The characteristic of MC is unique nuclear changes, such as the appearance of multinucleated or micronucleated cells. Giant multinucleated cells arise from improperly separated uncondensed chromosome clusters, while micronucleated cells originate from lagging chromosomes or chromosome fragments in late stages of cell division. These chromosomes or chromosome fragments are left behind in the cell division end to form daughter nuclei, resulting in the formation of micronuclei in addition to the main nucleus (Imreh et al., 2016). From a functional perspective, MC can be defined as an intrinsic tumor suppression mechanism. After mitotic drive failure, cell proliferation levels decrease, ultimately leading to cell death or senescence. Therefore, inducing MC can inhibit tumor growth (Dominguez-Brauer et al., 2015; Komlodi-Pasztor, Sackett & Fojo, 2012). The role of MCRGs in tumors has been confirmed. SENP3 is an important gene that regulates the cell cycle, playing a key role in the correct separation process of sister chromatids during mitosis. A previous study (Hu et al., 2022) reported that the activation of SENP3 in tumor cells could be coupled with the cGAS signaling, synergistically promoting host anti-tumor immune responses. The lack of SENP3 can lead to an increase of AKT1 SUMOylation, thereby regulating the polarization of macrophages towards the M2 subtype, promoting the progression of breast cancer (Xiao et al., 2022). Aurora kinase is a class of serine/threonine protein kinases that are most important in the process of mitosis, and they can regulate mitosis-related proteins by controlling phosphorylation. Researchers (Chen et al., 2024) found that Aurora kinase A (AURKA) induced apoptosis and ferroptosis in Ewing’s sarcoma cells by inhibiting the NPM1/YAP1 axis, suggesting that AURKA may be a potential target for Ewing’s sarcoma. In addition, in skin cutaneous melanoma, AURKA can inhibit the infiltration of CD8+ T cells and promote hypoxia by activating the TGF-β signaling pathway. Researchers believed that AURKA could regulate the infiltration levels of various immune cells in the tumor microenvironment, reshape the immunosuppressive tumor microenvironment, regulate cell apoptosis and hypoxia, and serve as a prognostic biomarker and potential therapeutic target for skin cutaneous melanoma (Long & Zhang, 2023). The cell cycle of mammals is regulated by a series of cell cycle checkpoints. CHK1 and CHK2 are important cell cycle checkpoint kinases that play a crucial role in maintaining the integrity of the cell genome. In multiple myeloma cells, CHK1 can interact with the STAT3 pathway. Inhibiting CHK1 can effectively suppress STAT3 tyrosine phosphorylation and DNA binding activity, thereby blocking the activation of downstream targets (Zhou et al., 2022). In HER2-positive gastric cancer cells, inhibiting CHK1 phosphorylation can enhance the sensitivity of cancer cells to lapatinib. This is manifested by downregulation of phosphorylated AKT and ERK, exacerbation of DNA damage, and enhancement of anti-proliferative effects (Bai et al., 2018). CHK2 can promote the progression of liver cancer by enhancing chromosomal instability, tumor heterogeneity, drug resistance, and immune evasion (Carloni et al., 2018). In prostate cancer, high CHK2 expression is associated with adverse tumor characteristics and can independently predict early tumor recurrence (Eichenauer et al., 2020).

Due to the important role of MC in tumor development, this study comprehensively analyzed the relationship between MCRGs and breast cancer using bioinformatics technology, in order to provide new directions for subsequent basic research. During the research process, we identified 194 MCRGs that were differentially expressed in breast cancer. GO and KEGG enrichment analysis results showed that differentially expressed genes were mainly enriched in cell cycle, mitosis, and chromosome segregation. Through Lasso regression analysis and Cox regression analysis, we ultimately identified four MCRGs and constructed a prognostic risk model. We divided patients into high-risk and low-risk groups based on the median risk score. We used ROC curve to validate the predictive ability of the model, and plotted a nomogram for predicting the prognosis of breast cancer patients in conjunction with clinical parameters. According to the results of the multivariable Cox regression analysis, this risk model was an independent factor affecting the prognosis of breast cancer patients. The GSEA analysis results between high-risk and low-risk groups showed that the high-risk group was significantly associated with base excision repair, cell cycle, DNA replication, and oxidative phosphorylation. Subsequently, we explored the relationship between risk scores and tumor immune microenvironment. The low-risk group showed abundant immune cell infiltration and more non-tumor cell components. In addition, the low-risk group had more active immune function and higher levels of immune checkpoint activation, which corresponded to better survival outcomes. TMB analysis showed that TMB affected patient prognosis, with higher levels of TMB associated with poorer OS. We also explored the sensitivity of high-risk and low-risk group patients to immunotherapy and chemotherapy drugs, providing reference for the treatment of breast cancer patients. Finally, we validated the gene expression of four MCRGs involved in constructing the prognostic model at the cellular and tissue levels through RT-qPCR experiments.

Four MCRGs involved in constructing the prognostic model play a crucial role in the development of tumors. PLK1 plays an important role in cell cycle regulation, with its main functions being to regulate entry into mitosis, centrosome maturation, formation and stability of the spindle poles, and cytokinesis (Kishi et al., 2009). In esophageal squamous cell carcinoma, downregulation of PLK1 can inhibit the pentose phosphate pathway, reduce NADPH and GSH levels, thereby promoting ferroptosis, and increasing the sensitivity of cancer cells to radiotherapy and chemotherapy (Zhao et al., 2023). S100B encodes calcium-binding protein, which plays an important role in mediating inter-tissue inflammatory responses and cell differentiation, invasion, and transformation in multiple cell cycle processes (Camidge, Doebele & Kerr, 2019). Research (Jiang et al., 2011) suggested that downregulation of S100B expression in non-small cell lung cancer (NSCLC) could significantly inhibit cell cycle progression, reduce colony formation, cell migration, and invasion activities, leading to decreased cell proliferation. Their study indicated that knocking down S100B may be a potential strategy for treating brain metastasis of NSCLC with S100B overexpression. IRS2 can regulate glucose metabolism and cell cycle (Manohar et al., 2020). Studies reported that it promoted the development of pancreatic cancer, renal cancer, and oral cancer by mediating the IGF1 and AKT signaling pathways (Gao et al., 2014; Ma et al., 2015; Stoeltzing et al., 2007). Interferon- γ encoded by the IFNG is a cytokine that can enhance the activity of macrophages and T lymphocytes, and kill tumor cells through the body’s immune mechanisms (Kursunel & Esendagli, 2016). MCRGs have been confirmed to have regulatory effects in various tumors, providing us with a new direction for further research on MC and breast cancer.

However, this study still has certain limitations. First, the predictive model was based on clinical information and gene expression information from public databases, and the validation method for the effectiveness of the prognostic model was relatively single, requiring more clinical data to verify its effectiveness. Second, there had been no experimental research on the tumor mechanisms of the genes included in the model, and it was still unclear how these genes affect tumor proliferation, metastasis, and prognosis. Next, we will conduct more basic research to further explore the relationship between the genes included in the model and MC, as well as the impact of these genes on the occurrence and development of breast cancer.

Conclusions

We constructed a prognostic model for breast cancer patients based on MCRGs for the first time, and further explored the relationship between MCRGs and immune infiltration in breast cancer, which may provide new targets for breast cancer treatment. However, further research is needed on the relationship between the genes included in the model and breast cancer.

Supplemental Information

Supplemental Information 1 The original code data of breast cancer

Supplemental Information 2 The original gene expression data of breast cancer

Supplemental Information 3 Methods for RT-qPCR Analysis

Supplemental Information 4 MIQE checklist

Supplemental Information 5 Mitotic catastrophe-related gene list

Supplemental Information 6 Primer sequences of 4 model genes

Supplemental Information 7 Quantification Cq Results

We are very grateful for the data provided by databases such as TCGA, GEO, and HPA. Thanks to the reviewers and editors for their sincere comments.

Additional Information and Declarations

Competing Interests

Author Contributions

Human Ethics

Data Availability

The authors declare there are no competing interests.

Shuai Wang conceived and designed the experiments, performed the experiments, authored or reviewed drafts of the article, and approved the final draft.

Haoyi Zi performed the experiments, authored or reviewed drafts of the article, and approved the final draft.

Mengxuan Li analyzed the data, authored or reviewed drafts of the article, and approved the final draft.

Jing Kong analyzed the data, authored or reviewed drafts of the article, and approved the final draft.

Cong Fan analyzed the data, prepared figures and/or tables, and approved the final draft.

Yujie Bai analyzed the data, prepared figures and/or tables, and approved the final draft.

Jianing Sun analyzed the data, prepared figures and/or tables, and approved the final draft.

Ting Wang conceived and designed the experiments, authored or reviewed drafts of the article, and approved the final draft.

The following information was supplied relating to ethical approvals (i.e., approving body and any reference numbers):

The First Affiliated Hospital of Air Force Medical University gave the study its approval (KY20232266-C-1).

The following information was supplied regarding data availability:

The original code and gene expression data of breast cancer are available in the Supplemental Files.

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
