# Peer review of "Development and validation of a mitotic catastrophe-related genes prognostic model for breast cancer"

_PeerJ, doi:10.7717/peerj.18075_

## Round 0.1 · original submission · Minor Revisions

· Academic Editor

Minor Revisions

Please follow the reviewers' comment to improve figure readability, details about the experimental design, references and detailed discussion.

·

Basic reporting

Overview
This paper is very well written, with clear and concise writing. You clearly laid out the research gap, field, and background information. The English used is of high quality. The figures used are clear and well made. I have some minor points, but overall am very happy with the article.

Supplemental data error
A) There may be an error in the in supplemental table Table_S2. When opening the file, I received the error that the file format and extension of the file does not match. Opening it showed a small table with two columns: “Prime name” and “Sequence”, as well as two empty sheets. Is this correct? You may need to upload a new version of the file to avoid any file corruption issues.

Figure changes
B) In figure 9C, the y axis is labelled as “tumor tmbation burden”. I presume that is meant to be “tumor mutation burden”?


Minor text points
C) Throughout the whole paper, there are no spaces before in-text citations.
D) Lines 311-312: Change “Previous study…” to “A previous study”.

Experimental design

No comment.

Validity of the findings

No comment.

Additional comments

This article is clear and well presented. I have a few minor points and questions that are not essential for publication, but which may help readability of the paper, or help explain the results better.

Predictive results
A) On lines 215-216, you state that your survival analysis had strong predictive ability with AUC results of 0.711, 0.660, and 0.632. Would those scores be high enough to be considered strong for models in this field?

Question about TMB and OS
B) In Figure 9, you show that a high-TMB leads to a poorer OS rate. However, in 9E, it appears like there is a split between patients that are both high-risk and have high TMB, compared to all others (high risk/high-TMB, low risk/high TMB, low risk/low TMB), which appear to be grouped closer together. Could you explain that phenomenon?

Figure Readability – just some minor points that may improve the readability of the figures.
C) In Figure 9 (A and B), you show the top genes in low-risk and high-risk groups, respectively. However, you put the high risk figure first, rather than the low risk figure, like you do with the other figures. Changing the order around may improve readability of the figure and make your figures more consistent.
D) In Figure 4, adding a label to highlight that sections A-D are from the training set, and that E-H are from the validation set would improve readability.

Point of description of mitotic catastrophe
E) On line 70-71, you say that mitotic catastrophe is a new form of cell death, though the phenomenon has been known of since 1987 (though I understand some other recent papers also describe it as "novel"). Perhaps it would be better to describe it another way, such as “MC is a form of cell death…”. Otherwise, if the ‘new’ part of the statement is referring to a specific type of MC or its artificial induction, please make that clearer.

Reviewer 2 ·

Basic reporting

The manuscript developed the mitotic catastrophe risk score model to predict breast cancer survival. Generally, this manuscript adopted appropriate methods, and the results are logical and can answer the hypothesis proposed by the author. The language is generally precise and intelligible. Nevertheless, there are still some issues need to be modified:

The section of INTRODUCTION adopts a funnel structure, gradually focusing and guiding to the research aim. However, the discussion on the role of Mitotic catastrophe in tumorigenesis, development, treatment, and prognosis is not in-depth enough. Detailly:

1) Line 70 - 71: "MC is a new form of cell death that plays an important role in the development of tumors." Please provide the reference(s) for this sentence. Moreover, in what aspects does MC affect the development of tumors? What is the mechanism? Please describe it according to previous studies.

2) Line 78 - 79: "However, to date, there has been no research on constructing a prognostic model based on MC-related genes (MCRGs) in breast cancer". What is the relationship between MC-related genes and prognosis in other cancers? Please list several studies.

3) Line 57 - 58: For the incidence and mortality data of breast cancer, please cite the latest literature: Bray F, Laversanne M, Sung H, Ferlay J, Siegel RL, Soerjomataram I, Jemal A. Global cancer statistics 2022: GLOBOCAN estimates of incidence and mortality worldwide for 36 cancers in 185 countries. CA Cancer J Clin. 2024 May-Jun;74(3):229-263.

Experimental design

1) Line 100: "The filtering criteria were |logFC| > 1 and FDR < 0.05". Usually, for filtering criteria, the value of |logFC| should be no less than 2 or 1.5. Is it too loose to choose >1 as criteria in this manuscript?

2) The manuscript only used one data set (TCGA-BRCA) to screen differential genes. Usually, at least 2 datasets are used and the intersection is taken as the result.

3) Line 102 – 103, Line 193, Figure 2: The method and result should indicate which type of GO was analyzed? biological process, cellular component, or molecular function?

4) After the MC risk score model was constructed, the manuscript also performed Cox regression analysis using the MC risk score model and other clinical factors, and found that the MC risk score is an independent prognostic factor, which nicely excludes the result bias that may be caused by the imbalance of other characteristics between the low and the high MC risk score group. However, the manuscript did not include the factor of molecular subtypes, which is known to be strongly related to prognosis.

Validity of the findings

1) In the section of RESULTS, I suggest that the authors place the part of "TMB analysis" immediately after the part of "Immune landscape", since TMB and immune landscape are both related to the immune microenvironment and immunotherapy, and then present the content related to treatment (the part of "Immunotherapy and drug sensitivity analysis").

2) I recommend to clearly state the expression level (up-regulated or down-regulated) of the 4 prognosis-related genes (PLK1, S100B, IRS2, IFNG) in the part of “Construction and validation of MCRGs-related prognostic model”, so as to correspond to the subsequent immunohistochemical staining results (the last part of RESULTS).

3) Figure 10: please provide patient ID for each image.

Additional comments

1) Line 173 – 174: "We obtained verbal consent from the patients". I am not sure if verbal consent instead of written consent meets the ethical standards?

2) The manuscript analyzed the prognostic factor of overall survival. In the future, if the recurrence data is available, I recommend the authors also to analyze the relationship between MC risk score and recurrence, which could further illustrate the application value of the developed MC risk score model.

---

## Round 0.2 · accepted · Accept

· Academic Editor

Accept

Great revision, ready to publish.

·

Basic reporting

No comment

Experimental design

No comment

Validity of the findings

No comment

Additional comments

This work has been updated well, and all previous comments have been addressed. This is a well written and communicated work. Thank you.

One *VERY* minor point I noticed was in the abstract, line 47: ... prognostic associated MCRGs...
Typically this is hyphenated (e.g., prognostic-associated MCRGs...). This does not need to be addressed for publication, just a small note.

Thank you for your updates and work.

Reviewer 2 ·

Basic reporting

The authors have provided responses and carried out revisions in accordance with the majority of the comments. The responses have addressed the questions I raised to a considerable extent, and the manuscript has been enhanced. Nevertheless, regarding two of the issues: The filtering criteria of |logFC| > 1 is too loose, and only one data set was used to screen differential genes, the authors did not made revisions but expounded their opinions,I reckon these opinions can be grudgingly accepted. No further comments.

Experimental design

N/A

Validity of the findings

N/A

Additional comments

N/A